# The Risk of Autoimmunity Development following mRNA COVID-19 Vaccination

**DOI:** 10.3390/v14122655

**Published:** 2022-11-28

**Authors:** Jerzy Świerkot, Marta Madej, Magdalena Szmyrka, Lucyna Korman, Renata Sokolik, Iga Andrasiak, Ewa Morgiel, Agata Sebastian

**Affiliations:** 1Department of Rheumatology and Internal Medicine, Wroclaw Medical University, Borowska 213, 50-566 Wroclaw, Poland; 2WroMedica Research Center, 50-566 Wroclaw, Poland

**Keywords:** COVID-19, vaccination, autoantibodies, autoimmune diseases

## Abstract

The broad spectrum of interactions between autoimmune diseases and the SARS-CoV-2 vaccination is not fully understood. This study aims to evaluate the prevalence of anti-nuclear antibodies (ANA), anti-ENA, anticardiolipin antibodies (ACL), and anti-beta-2 glycoprotein I antibodies (anti-β2GPI) before and after the SARS-CoV-2 mRNA vaccination in a real-life setting in healthcare professionals. The identification of risk factors associated with vaccine immunogenicity was evaluated. The study group consisted of employees of two hospitals (354 individuals). Samples for antibody assays were collected before vaccination and at 7–9 months after complete immunisation. There was no significant increase in the prevalence of ANA, ACL or anti-β2GPI antibodies, or autoimmune diseases in subjects who were vaccinated 7–9 months after complete immunisation. In terms of detected anti-ENA, the anti-DFS70 antibodies were found in 6 times more subjects than before vaccination at the second blood draw (in 18 and 3 subjects, respectively) (*p* = 0.001). There were no significant relationships between a SARS-CoV-2 infection history, humoral response, cellular response, subject category, smoking, sex, body weight, ANA, anti-ENA, ACL, or anti-β2GPI. This study revealed a possible association between the severity of vaccine adverse events (VAEs) and ANA titre. Individuals with more severe VAEs (>10 points) after the second dose of the vaccine had significantly higher ANA titre after complete immunization. When analysing the significance of time between the ANA, anti-ENA, ACL, and anti- β2GPI assays and complete immunisation antibody values, no qualitative result was statistically significant. There was correlation between the time since complete immunization and ANA after.

## 1. Introduction

The global SARS-CoV-2 pandemic has already caused more than 477 million infections and 6.1 million deaths worldwide and contributed to impeded access to health care, as well as severe economic, sociological and psychological damage. 

Vaccination is the most promising way to reduce the morbidity and mortality associated with SARS-CoV-2 infection. To date, phase III clinical trial results have revealed that both Pfizer/BioNTech (BNT162b2) and Moderna (mRNA 1273) mRNA vaccines achieved 90–95% efficacy in protecting against severe COVID-19 with a very favourable safety profile [1]. 

A vaccine needs a pathogen-specific immunogen and an adjuvant to stimulate acquired immunity. An optimal adjuvant stimulates innate immunity without inducing systemic inflammation that could cause serious adverse effects. For mRNA vaccines, mRNA can serve as both an immunogen (encoding a viral protein) and adjuvant, due to the intrinsic immunostimulatory properties of RNA. The mRNA vaccines require two doses 3–4 weeks apart for optimal protection. They are often associated with mild-to-moderate vaccine adverse events (VAE), including injection-site pain, transient fever, and chills that may be more severe after the second vaccination.

The broad spectrum of interactions between autoimmune diseases and SARS-CoV-2 vaccination is not fully understood. The activation of the interferon pathway is one of the mechanisms of action of mRNA vaccines against SARS-CoV-2. Various autoimmune diseases are increasingly described in the literature, after natural infection with the SARS-CoV-2 virus and SARS-CoV-2 vaccination [2]. However, significant autoimmune disorders caused by mRNA vaccines have not been clearly identified to date. The potential induction of antinuclear antibodies (ANA), specific extractable nuclear antigens (anti-ENA), and antiphospholipid antibodies (APLA) following infection and vaccination has also been of interest to researchers. Some studies revealed a higher prevalence of autoantibodies in COVID-19 patients [3].

A small number of people vaccinated against SARS-CoV-2 infection develop side effects, including autoimmune syndromes. These include venous thrombosis and thrombocytopenia within a few days (7–10 days) of the ChAdOx1 (Astra Zeneca) vaccination. This syndrome is called VITT, vaccine-induced immune thrombotic thrombocytopenia. Patients have elevated titres of antibodies binding to complexes of platelet factor 4 and polyanions. These antibodies develop in patients who have not been exposed to heparin before. Other autoimmune syndromes have also been described, such as severe autoimmune thrombocytopenia with anti-Ro/SSA antibodies and lowered levels of complement components [4].

This study aims to evaluate the immunogenicity of the mRNA vaccination by assessing the prevalence of ANA, anti-ACL, and anti-β2GPI antibodies before and after complete basic immunization with the mRNA vaccine against SARS-CoV-2 in a real-life setting in healthcare professionals. This study also includes the analysis of the potential association of post-vaccination response with immune response in the form of autoantibody synthesis and the onset of autoimmune diseases between 7–9 months after the completion of basic immunization.

## 2. Materials and Methods

### 2.1. Study Group and Study Design

The study group consisted of medical professionals who received a complete COVID-19 immunization and had blood assays done at three scheduled time points (before vaccination, before the second dose of vaccine, and 7–9 months after the first vaccination). Eligibility criteria for the study were age ≥ 18, vaccination against COVID-19, active employment in the hospital, and signature consent to the study. The criterion for exclusion from the study was a failure to obtain written consent for the study. The healthcare professionals were divided according to the nature of their work into the following categories: physicians, nurses/paramedics, physiotherapists, care managers, room attendants, administrative staff, and laboratory staff. All participants were vaccinated with Pfizer-BioNTech’s BNT162b2. The first dose was administered from December 2020 to February 2021, while the second dose was from January to March 2021. The median interval between vaccinations was 21 (IQR:21–21) days. Assays of antibodies such as ANA, anti-ENA, ACL, and anti-β2GPI were performed before vaccination (ANA T0; anti-ENA T0, APLA T0) and before the third dose in August–September 2021 (ANA T1; anti-ENA T1, APLA T1) (median number of days: 230, IQR: 224–241.5).

The kinetics of the post-vaccination response was examined by the titres of anti-SARS-CoV-2 IgG antibodies against the S1 protein (S1 subunit of the S protein) located within the virus spike in blood samples collected on three occasions. Antibody levels before vaccination were determined as anti-SARS-CoV-2 IgGT0, 4–9 weeks after complete immunization as anti-SARS-CoV-2 IgGT1, and approximately 7–9 months after vaccination (before the booster dose) as anti-SARS-CoV-2IgGT2. The cellular response was tested once in a group of employees of one hospital (*n* = 189) at the third blood draw using the Quan-T-Cell SARS-CoV-2 IGRA EUROIMMUN assay.

### 2.2. Assays

Humoral immune responses were measured with a EUROIMMUN anti-SARS-CoV-2 QuantiVac ELISA (IgG), which allows precise quantitative testing of IgG-class neutralizing antibodies directed against the S1 protein of the SARS-CoV-2 virus (compatibility with neutralisation tests). According to the manufacturer’s recommendations, antibody levels above 35.2 BAU/mL are considered a positive result.

Cellular immune response measurements were performed using a EUROIMMUN QuanT-Cell SARS-CoV-2 (IGRA) assay. According to the manufacturer’s recommendations, antibody levels above 200 mIU/mL are considered a positive result.

ANA antibodies were performed on a mosaic substrate HEp2 cells/primate liver manufactured by EUROIMMUN Medizinische Labordiagnostika AG. The indirect immunofluorescence test (IIFT) was used. ANA-positive samples (≥1:100) were further tested to determine antibody titres. Specific antibodies to anti-ENA were determined using the dot-blot technique. Assays were performed using EUROIMMUN reagents—EUROLINE test using 17 antigens: Sm/nRNP, Sm, Ro-52, Ro60/SS-A, La/SS-B, Scl-70, PM-Scl, Jo-1, CENP-B, PCNA, dsDNA, nucleosomes, histones, RibP, AMA-M2, Dense Fine Speckled, 70 kDa (DFS70) performed on EuroBlotMaster.

Antiphospholipid antibodies: assays of ACL and anti-β2GPI antibodies in IgG and IgM classes were performed with ELISA, using EUROIMMUN commercial kits. These tests were performed on ANALYZER I. In terms of ACL antibodies, titres > 12 U/mL were considered positive results according to the manufacturer’s instructions, while in terms of anti-β2GPI IgG/IgM antibodies—above 20 RU/mL.

Data concerning health status and lifestyle were based on surveys conducted among the subjects and performed before vaccination and the third dose in August–September 2021. Side effects after the first and second vaccination dose were assessed in separate surveys, including the occurrence of local reactions (i.e., redness, swelling, pain), general reactions (fever, fatigue, headache, chills, vomiting, diarrhoea, myalgia, arthralgia), and grading of severity on a scale of 0—none, 1—mild, 2—moderate 3—severe, 4—very severe based on Food and Drug Administration guidance for toxicity grading scales for vaccines [5].

This study received approval from an independent ethics committee (No. KB 634/2020) and fulfilled the ethical guidelines of the Declaration of Helsinki. All study participants provided written informed consent before enrolment.

### 2.3. Statistical Analysis

Statistical analysis was performed using the R Project for Statistical Computing, version R 4.1.1, Vienna, Austria. Categorical variables were shown as frequencies and percentages, whereas basic descriptive statistics (minimum, maximum, interquartile range (IQR), median, mean, and standard deviation (SD) or standard error (SE)) were used for describing continuous variables. Evaluation of data normality was performed using the Shapiro-Wilk test. Categorical variables were compared using the χ2 test and Fisher’s exact test. For paired nominal data, the McNemar’s test was applied. Continuous variables were compared using the Spearman’s rank correlation coefficient. The Mann–Whitney U test was used for comparing categorical and not normally distributed continuous variables. For multiple comparisons, the Kruskal-Wallis test and Dunn’s post hoc test with Benjamini–Hochberg correction were applied. A *p*-value < 0.05 was considered statistically significant.

## 3. Results

### 3.1. Study Group

The study population consisted of 354 employees from University hospital, who received complete basic COVID-19 immunisation (Pfizer-BioNTech’s BNT162b2) and from whom three blood samples were collected at intervals: before vaccination and 7–9 months after primary immunisation.

The mean age of the subjects was 49 ± 11 years and median age was 50 years (IQR: 43–57). The youngest subject was 22, and the oldest was 72. Women represented 82% (*n* = 289) of the study population. In subjects, the distribution by the professional group was as follows: nurses/carers/paramedics 43% (*n* = 152), non-surgeons 21% (*n* = 74), administrative staff 9% (*n* = 33), laboratory staff 15% (*n* = 55), surgical physicians 6% (*n* = 19), physiotherapists 3% (*n* = 10), ward attendants, and stretcher-bearers 3% (*n* = 11).

One hundred sixty-one individuals who underwent infection before vaccination were identified by a positive nasopharyngeal swab using a PCR test and/or a positive anti-SARS-CoV-2 IgG or IgM serologic test performed before COVID-19 vaccination. In the study group, 54% of subjects (193/354) had not experienced SARS-CoV-2 infection. Chronic diseases were reported in 111 out of 354 (31%) participants. Before the beginning of the study, 47 patients were diagnosed with autoimmune and allergic diseases. Detailed characteristics of the study group are shown in Table 1.

### 3.2. Changes in ANA, Anti-ENA, ACL, and Anti-β2GPI Profile before and after SARS-CoV-2 Vaccination

ANA, anti-ENA, ACL, and anti-β2GPI antibodies were performed twice: before vaccination (T0) and 7–9 months after basic immunisation (T1) (Table 2).

In the study population, positive ANA antibody titres were found in 161 (45%) subjects at the first blood draw (before SARS-CoV-2 vaccination) and 165 (47%) at the second blood draw 7–9 months after primary immunisation.

In 68 subjects, ANA antibody titres decreased to 0 by the second measurement, while 74 subjects who initially did not have ANA antibodies proved to be positive (64 of them had a titre of 1:100). When analysing only the population with ANA antibody titres of 1:320 (*n* = 39) and higher at follow-up after 7–9 months, there was an increase in the titres in only five subjects, while there was no change in the titres in the other 13, and 21 people experienced a decrease in their ANA antibody titres (Table 3).

All subjects with DFS70 or PmScl at 7–9 months had ANA T1 titres of at least 1:320. DFS70 antibodies at the second blood draw were found in 6 times more subjects than before vaccination (*p* = 0.001). In contrast, PmScl antibodies, which were found in only one subject before vaccination, were found in five subjects at follow-up after 7–9 months (differences were not statistically significant).

At the first blood draw (before SARS-CoV-2 vaccination), positive ACL IgG T0 antibody titres were found in 2 (0.5%) subjects, ACL IgM T0 in 4 (1%), anti-β2GPI IgM T0 in 26 (7%), and none in anti-β2GPI IgG T0.

At the second blood draw 7–9 months after complete immunisation, positive ACL IgG T1 antibody titres were found in 5 (1.4%) subjects, ACL IgM T1 in 6 (1.7%), anti-β2GPI IgM T1 in 17 (5%), and none in anti-β2GPI IgG T1. Positive ACL IgM T0 and ACL IgM T1 antibodies were found only in women older than 50 years. Four women with ACL IgM T1 were found to be ANA T1 positive (two with titres above 1:320). Out of 6 subjects with positive ACL IgM T1, three already had positive ACL IgM T0 before vaccination.

In the case of anti-β2GPI IgM T1 positive subjects (*n* = 17), 82% were women, and most of them (16 out of 17) were found to be anti-β2GPI IgM T0 positive before vaccination. Seven to nine months after complete immunisation, anti-β2GPI IgM was found in 35% fewer subjects (before in 26, after in 17) than before vaccination.

The obtained data showed no increase in the prevalence of autoimmune diseases in those vaccinated 7–9 months after complete immunisation.

### 3.3. The Presence of ANA, Anti-ENA, ACL, Anti-β2GPI Antibodies According to Sex, Weight and Age

In the study group, there were no statistically significant differences in the prevalence of ANA T0/ANA T1, anti-ENA T0 and T1, ACL IgG/IgM T0 and T1, respectively, or anti-β2GPI IgG/IgM T0 or T1 according to sex and body weight.

The relationship between the age and presence of tested antibodies was also analysed. There was a relationship between age and anti-β2GPI IgM T0 (*p* = 0.0022)/age and ACL IgM T1 (*p* = 0.043) levels (Table 4). The performed tests did not show the presence of anti-β2GPI IgG antibodies (T0 and T1). Therefore, they were not included in the analysis in Table 4.

The median age for those with negative anti-β2GPI IgM T0 antibodies was 49 (IQR 43–56) and positive anti-β2GPI IgM T0 antibodies 57 (IQR 51–61). There was an association between age and anti-β2GPI IgM T1 value (Figure 1).

There was no age, sex, or comorbidities-related increase in the prevalence of positive ANA T0 and ANA T1 antibodies or anti-ENA T0 and anti-ENA T1.

ANA T0 antibodies in subjects aged > 50 years at titres above 1:100 were found in 42% of the subjects, at titres above 1:320 in 10%, while in those aged 50 and under—48% and 11%, respectively. ANA T1 antibodies in subjects aged > 50 years at titres above 1:100 were found in 47% of the subjects, at titres above 1:320 in 10%, while in those aged 50 and under—47% and 16%, respectively.

### 3.4. The Presence of ANA, Anti-ENA, ACL IgG/IgM, Anti-β2GPI IgG/IgM According to the Time Elapsed since Complete Immunisation

When analysing the significance of time between the 2nd ANA, anti-ENA, APLA assays and complete immunisation (7–9 months after complete immunisation)/antibody values no qualitative result was statistically significant.

There was a relationship between time since complete immunisation and the ANA T1 value (Figure 2).

This study also analysed the relationship between IgG T0, IgG T1, IgG T2 anti-SARS CoV-2 antibody levels and ANA, anti-ENA, ACL, and anti-β2GPI antibodies. No quantitative or qualitative relationships were found. There were also no significant quantitative or qualitative relationships between cellular responses determined by QuanT-Cell and ANA, anti-ENA, ACL, or anti-β2GPI antibodies.

Similarly, when comparing the prevalence of positive ANA, anti-ENA, ACL, and anti-β2GPI antibody results in different professional groups studied, no statistically significant differences were revealed. There was also no relationship between the presence of antibodies and smoking. Also the titres of ANA, anti-ENA, ACL, and anti-β2GPI antibodies were not affected by the SARS-CoV-2 infection history (before vaccination).

### 3.5. The Analysis of Reported VAEs and Presence of ANA, Anti-ENA, ACL, and Anti-β2GPI Antibodies

In the analysed population, the prevalence of VAE1 (after the first dose of the vaccine) with a severity of 0–5 points, 6–10 points, and >10 points was 67%, 17%, and 16%, respectively, while VAE2 (after the second dose of the vaccine) was 52%, 24%, and 24%, respectively. The median severity of VAE1 was 3 (IQR: 2.00–7.00) and after the second dose (VAE2) 5 (IQR: 2.00–10.00).

There was no relationship between VAE1 and ANA T0, ANA T1, ACL IgG T0/IgM T0, ACL IgG T1/IgM T1, anti-β2GPI IgG T0/IgM T0, or anti-β2GPI IgG T1/IgM T1 values (quantitative and qualitative evaluation).

There was a relationship between the presence of VAE2 and ANA T1 titres. There were differences between VAE2 0–5 points/VAE2 >10 points and ANA T1 titres of 66.41 (±120.15) (SE = 8.9) vs. 139.29 (±235.57) (SE = 25.7), respectively (*p* = 0.0188) (Table 5). The qualitative evaluation revealed no statistically significant relationships between VAE2 and the analysed variables—ANA, ACL, anti-β2GPI.

### 3.6. Chronic Diseases and the Presence of ANA, Anti-ENA, ACL IgG/IgM, Anti-β2GPI IgG/IgM Antibodies

The comparison of quantitative variables by total chronic disease prevalence revealed no statistically significant differences. We also analysed qualitative changes due to the presence of chronic diseases. The results were statistically significant for ACL IgG T1; the relationship between the occurrence of chronic diseases and ACL IgG T1(*p* = 0.0367) was revealed. In patients with a positive ACL IgG T1, chronic diseases were reported 4 times more often (80% vs. 20%), whereas in patients with a negative ACL IgG T1, chronic diseases were reported more than 2 times less frequently (31% vs. 69%).

When analysing chronic diseases individually, i.e., cardiovascular diseases, lung diseases, neurological diseases, Hashimoto’s disease, diabetes mellitus, there were no quantitative or qualitative changes with respect to ANA, anti-ENA, ACL, or anti-β2GPI antibodies.

## 4. Discussion

Observational studies, single or case series reports, suggest the relationship between SARS-CoV-2 infection and the autoimmunity/synthesis of ANA [6] and APLA [7] antibodies. A considerable amount of data shows the risk of autoimmune diseases following a history of COVID-19 infection [8,9]. However, limited knowledge exists about the possible relationship between the COVID-19 vaccination and synthesis of autoantibodies. In this study, the authors evaluated selected autoantibodies in a population of employees from two hospitals before and approximately 7–9 months after complete BNT162b2 mRNA vaccination. According to the authors’ knowledge, no such analyses have been conducted to date. In the study group the presence of ANA before vaccination was reported in almost half of the subjects, while APLA was present in a few percent of subjects. When evaluated 7–9 months after complete immunisation, there was no significant increase in autoantibody titres or the presence of autoimmune disease.

Cases of autoimmune diseases with positive ANA tests have been reported in the literature, the occurrence of which temporally coincided with COVID vaccination with vaccines of different mechanisms of action (vector and mRNA) [2,10]. Ishay et al. presented a series of eight cases of de novo autoimmune diseases or exacerbations of previously diagnosed diseases in the setting of mRNA (BNT162b2 mRNA) vaccine exposure. However, in no case was there a correlation with the presence of ANA antibodies [11]. Neurological disorders, autoimmune syndromes, and thrombosis have also been described after administration of various types of vaccines [4,12,13]. According to the authors’ observations, none of the vaccinated individuals developed an autoimmune disease.

SARS-CoV-2 mRNA vaccines increase the levels of type I interferon (IFN), which is known to not only play a key role in the antiviral immune response but is also an important cytokine in the pathogenesis of systemic connective tissue diseases [14].

The phenomenon of post-vaccination autoimmune complications at the cellular level can be explained by the stimulation of type 1 IFN synthesis after vaccination exposure, which disrupts cellular tolerance mechanisms and induces the synthesis of ANA autoantibodies [2,15]. Theoretically, activation of the innate immune system may contribute to nonspecific activation of autoreactive lymphocytes, leading to a higher risk of recurrence of primary autoimmune disease or hypothesised development of de novo disease [16]. The mechanisms involved in vaccination and autoimmunity are still incompletely understood. The most likely process involved is the phenomenon of molecular mimicry. Protein S may share similarities with neurological, endocrine, gastrointestinal, and myocardial autoantigens. Cross-reactivity is determined by environmental factors and genetic predisposition, such as deficits in immune tolerance following abnormal presentation of major histocompatibility complex (MHC) class II antigens to autoreactive T cells [9,10]. The development of immune responses to epitopes distinct from the disease-causing epitope may be an alternative explanation for the autoimmune response associated with COVID-19 vaccines. The authors also take into account that vaccines may only be the initiator and not the cause of autoimmunity. It may be possible that in susceptible individuals, vaccines accelerate the development of overt autoimmunity by stimulating autoreactive T and B lymphocytes [9]. It is also evident that Th-2 skewed immune systems may not fare as well with exposure to SARS-CoV-2 epitopes in vaccines or exposure by infection [17]. It is likely that a single theory does not explain all cases of post-vaccine autoimmunity.

ANA antibody assays are one of the primary diagnostic tools used in the evaluation of autoimmune processes in daily clinical practice. ANA antibodies are found in many autoimmune diseases. Moreover, many investigators note the significant percentage of healthy individuals with the presence of ANA, although usually at low titres [18,19,20]. In the study group, ANA positivity was found in 45% of subjects before vaccination and in 47% of subjects after vaccination. Previous studies revealed the ANA positivity rate of 20–50% in healthy populations. Environmental factors associated with working in a hospital that may influence autoantibody induction should also be considered, such as exposure to microorganisms or high exposure to disinfectants. Interestingly, Marin et al., while determining ANA in healthy subjects also reported the highest rate of positivity among hospital personnel [21]. The large discrepancy in the percentage of positive results between populations is influenced by many factors, including the technique used for performing the assay, cut-off point adopted, and demographic differences [19,20,22]. In the authors’ group, the pre- and post-vaccination assays revealed no statistically significant differences in terms of either qualitative or quantitative assessment (antibody titre). However, it should be noted that the ANA status of a large group of subjects changed. The result changed from positive to negative ANA status in 68 subjects and from negative to positive in 74. Only 57/165 subjects with the ANA antibody present had the same titre in the next assay; most of these changes (87%) involved a change in ANA titres from negative to 1:100 and vice versa. In light of the high percentage of positive results for ANA antibodies, the role of ANA antibodies in the body should be discussed. Pashnina et al. presented an interesting concept concerning the functioning of the immune system, attributing certain physiological functions to ANA [23]. According to this theory, ANA as a bioregulator should be within a designated range of values where both its absence and excess may lead to a disease process. This rather controversial concept needs further scientific verification.

Some of the authors who analysed the age-related characteristics of ANA-positivity concluded that its prevalence increases with age. According to other researchers, no correlation was found between ANA titres and the age of adult donors, at least in the range of 20–60 years [21,23]. In our study group, there were only 10 people over the age of 65 (66–72 years old), and there was no participant under the age of 20.

The data concerning the role of anti-DFS70 antibodies are one of the important arguments for looking at autoantibodies in a slightly different way. Antibodies to the DFS70 antigen are found in healthy individuals and many chronic inflammatories and sometimes autoimmune conditions [24,25,26]. DFS70 antibodies are responsible for approximately 10% of ANA positivity by IFT in routine testing. Based on data available to date, anti-DFS70 antibodies do not appear to play a pathogenetic role in autoimmune diseases. There was as much as a 6-fold increase in the prevalence of anti-DFS70 antibodies in the analysed group of subjects after COVID-19 vaccination. This may be related to the role of DFS70 as a transcriptional regulator of a gene activated in response to an infectious agent that can cause oxidative stress [27]. To date, however, there is no evidence that modifications due to oxidative stress increase the immunogenicity of this protein. In a recently published study concerning DFS70 and oxidative stress, the authors even proved that there was reduced oxidative stress in samples positive for anti-DFS70 antibody, especially in samples with high antibody titres. Therefore, anti-DFS70 antibodies can be considered an indirect marker of the body’s antioxidant response. In the case of the population analysed by the authors, vaccine administration may have been the triggering factor and the production of anti-DFS70 antibodies indirectly may be further evidence of their protective role [28]. This is especially true since none of the observed subjects developed systemic connective tissue disease. According to the authors’ knowledge, this study is the first to describe an increased incidence of DFS70 antibodies after COVID-19 vaccination.

An increased number of autoantibodies and at the same time a rare occurrence of autoimmune diseases are observed in older people. In a study conducted on COVID-19 patients, ANA antibodies were found in 4–50% of subjects, mainly among the elderly [29]. Thrombotic incidents were also more frequent in older patients; however, the association with the presence of antibodies was not clear [30]. In this study, there was no age-dependent increased prevalence of positive ANA or anti-ENA antibodies after complete immunisation. This may be because there were only 10 subjects (2.8%) aged over 65 in the study population. This study only proved that positive ACL antibodies after complete immunisation were found more often in the elderly; however, this was not associated with thromboembolic complications. Furthermore, the percentage of positive results was similar to that described in the healthy population.

Moreover, vaccines based on adenoviral vectors can bind platelets and induce their destruction in reticuloendothelial organs. Vaccines based on liposomal mRNA may instead promote the activation of clotting factors and confer a prothrombotic phenotype to endothelial cells and platelets. Moreover, both preparations may trigger an IFN I response associated with the generation of aPL. In turn, APLA can lead to aberrant immune response activation by innate immune cells, cytokines, and the complement cascade. NETosis, monocyte recruitment, and cytokine release may further support endothelial dysfunction and promote platelet aggregation. These considerations suggest that APLA may be a risk factor for thrombotic events following COVID-19 vaccination. Contrary to our hypothesis, we observed a statistically significant (2%) decrease in anti-β2GPI IgM antibodies after vaccination. However, due to the small number of participants with a positive titre of these antibodies, we are not able to draw specific conclusions from this observation [31].

There were also no significant quantitative or qualitative relationships between humoral/cellular response to vaccination and the presence of ANA, anti-ENA, ACL, or anti-β2GPI antibodies. This study, however, found a possible association between the severity of VAEs and ANA titres. Those with more severe VAEs (>10 points) after the second dose of vaccine had significantly higher ANA titres when assessed 7–9 months after complete immunisation. This fact may suggest that increased VAEs (VAE2 > 10 points) may be related to the stimulation of autoimmune processes in the form of autoantibody synthesis. However, none of the subjects developed symptomatic autoimmune disease 7–9 months after complete basic immunization. According to the current knowledge, this study is one of the first to analyse the relationship between the severity of VAEs after vaccination with mRNA vaccine and synthesis of ANA antibodies. Data obtained from randomised clinical trials with respect to the association between a COVID-19 infection history and adverse effects after vaccination were inconclusive [32]. In the group analysed in this study, there was no association between the presence of the tested antibodies and previous SARS-CoV-2 infection. In contrast, Blank et al. reported an increased rate of ANA seroconversion after vaccination in subjects with a SARS-CoV-2 infection history compared with those who were not infected [33].

Our work was limited by the data being from only one study site, which may reduce their use in other populations.

## 5. Conclusions

The results of this study did not reveal any significant effects of a history of COVID-19 vaccination (Pfizer-BioNTech’s BNT162b2) on the presence of ANA, ACL, or anti-β2GPI antibodies, either in qualitative and quantitative evaluations. There was a weak positive correlation between time since complete immunisation and ANA value after 7–9 months. Even if a subject tests positive for ANA after immunisation, the pathogenic potential of these autoantibodies, their clinical significance, and how long they persist after vaccination are still unclear.

Multi-year studies that would definitively refute the hypothesis of the possible induction of autoimmune diseases by mRNA vaccines should still be conducted.

## Figures and Tables

**Figure 1 viruses-14-02655-f001:**
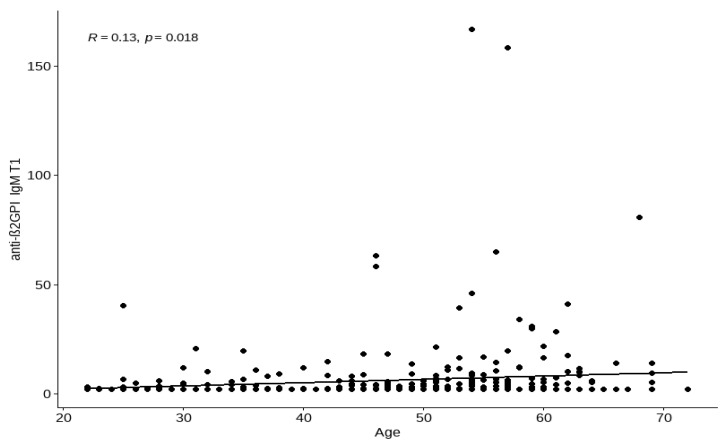
The quantitative analysis of the relationship between anti-β2-glycoprotein I antibodies (anti-β2GPI IgM T1) and the age of subjects before the third vaccine dose. The strength of these correlations is weak (R = 0.13). Abbreviations: T1-laboratory assay before the third dose (median number of days: 230).

**Figure 2 viruses-14-02655-f002:**
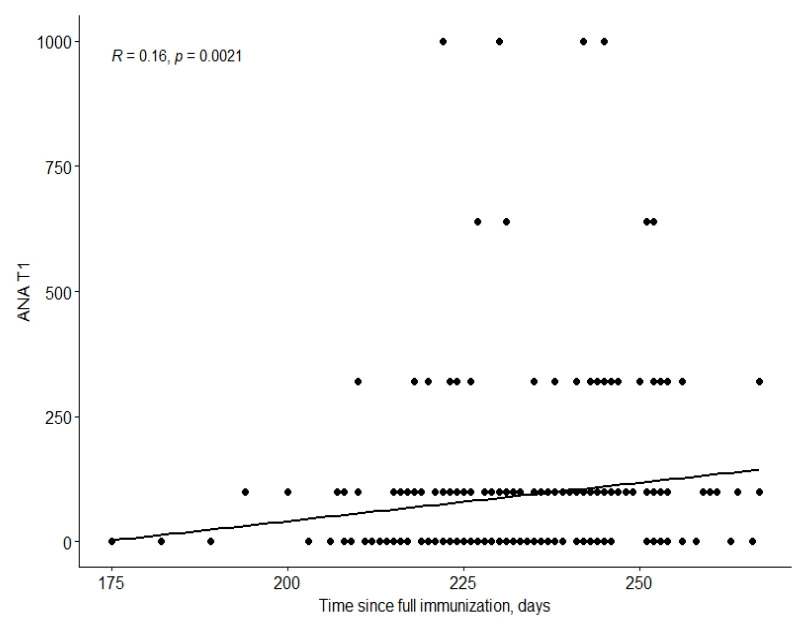
The relationship between time since complete immunisation and ANA values. The strength of these correlations is weak (R = 0.16). Abbreviations: ANA—antinuclear antibodies; T1—laboratory assay before the third dose (median number of days: 230).

**Table 1 viruses-14-02655-t001:** Characteristics of the study group of hospital employees.

	TOTAL, N (%)
**SEX:**	**354**
MALE	65 (18)
FEMALE	289 (82)
**PAST COVID-19 INFECTION**	161 (45)
**COURSE OF INFECTION:**	**158**
ASYMPTOMATIC	37 (23)
SYMPTOMATIC, HOME TREATMENT	111 (70)
SYMPTOMATIC, HOSPITALISATION	10 (7)
**SUBJECT CATEGORY**	
ADMINISTRATION	33 (9)
PHYSIOTHERAPIST	10 (3)
NON-SURGEON	74 (21)
SURGEON	19 (6)
NURSE/PARAMEDIC/CARER	152 (43)
LABORATORY ASSISTANT/PHARMACIST	55 (15)
WARD ATTENDANTS/STRETCHER-BEARERS	11 (3)
**SMOKING**	**350**
NO	292 (83)
YES	58 (17)
**BLOOD TYPE**	**319**
0	108 (34)
A	117 (37)
B	69 (21)
AB	25 (8)
**VAE 1**	
0–5	237 (67)
6–10	60 (17)
>10	57 (16)
**VAE 2**	
0–5	184 (52)
6–10	86 (24)
>10	84 (24)
**CHRONIC DISEASES:**	**111 (31)**
RENAL DISEASES	2 (0.6)
CARDIOVASCULAR DISEASES HYPERTENSION HYPERCHOLESTEROLAEMIA	36 (10) 11 (9.9) 2 (0.6)
LUNG DISEASES ASTHMA CHRONIC OBSTRUCTIVE PULMONARY DISEASE	6 (1.7) 3 (0.8) 3 (0.8)
CONNECTIVE TISSUE DISEASES	4 (1.1)
OSTEOPOROSIS	3 (0.8)
SKIN PSORIASIS	3 (0.8)
SARCOIDOSIS	1 (0.3)
NEUROLOGICAL DISEASES	5 (1.4)
HASHIMOTO’S DISEASE	31 (9)
THYROID NODULES	3 (0.8)
DIABETES MELLITUS	5 (1.4)
INSULIN RESISTANCE	2 (0.6)
GLAUCOMA	1 (0.3)
ALLERGY	7 (2.0)
ENDOMETRIOSIS	2 (0.6)

Abbreviations: VAE—vaccine adverse event; VAE1—vaccine adverse event after the first dose; VAE2—vaccine adverse event after the second dose; N—number of patients; %—percent of patients.

**Table 2 viruses-14-02655-t002:** ANA, ACL IgG/IgM, anti-β2GPI IgG/IgM assays in the study group, performed before vaccine administration and 7–9 months after basic immunisation.

Assessed Parameter	T0 N (%)	T1 N (%)	*p*-Value	Type of Statistical Test
ANA	161 (45)	165 (47)	0.797	McNemar’s
ACL IgG	2 (0.5)	5 (1.4)	0.371	McNemar’s
ACL IgM	4 (1)	6 (1.7)	0.617	McNemar’s
anti-β2GPI IgG	0 (0)	0 (0)	-	-
anti-β2GPI IgM	26 (7)	17 (5)	0.016	McNemar’s

Abbreviations: N—number of positive subjects; %—percent of patients; ANA—antinuclear antibodies; ACL—anti-cardiolipin antibodies; anti-β2GPI—anti-beta-2 glycoprotein I antibodies; T0—laboratory assay before vaccination; T1—laboratory assay before the third dose (median number of days: 230).

**Table 3 viruses-14-02655-t003:** ANA assays including titres in the study group before and after COVID-19 vaccination.

ANA Result	1:100	1:320	1:640	1:1000	1:3200	Positive Result	Negative Result
Before vaccination	122	32	3	3	1	161 (46%)	193 (54%)
7–9 months after vaccination	118	37	5	5	0	165 (47%)	189 (53%)

Abbreviations: N–number of positive subjects; %—percent of patients; ANA—antinuclear antibodies; Titre of ANA: 1: 100-320-640-1000-3200.

**Table 4 viruses-14-02655-t004:** The presence of anti-DFS antibodies, ACL, and anti-β2GPI antibodies according to age. Qualitative statistically significant results included age vs. ACL IgM T1 and age vs. anti-β2GPI IgM T0.

	Median (IQR) for Absence of Antibodies in Years	Median (IQR) for Presence of Antibodies in Years	*p*-Value of Mann-Whitney U Test
anti-DSF T0	57 (51–61)	54 (44–58.5)	0.6335
anti–DSF T1	47.5 (42.75–53.5)	48.5 (34.25–52.5)	0.7811
ACL IgG T0	50 (43–57)	59.5 (59.25–59.75)	0.0994
ACL IgG T1 ACL IgM T0	50 (43–57) 50 (43–57)	46 (46–52) 55 (53.75–56.25)	0.6492 0.2309
ACL IgM T1	50 (43–57)	56.5 (54.5–57)	0.0437 *
anti-β2GPI IgM T0	49 (43–56.5)	57 (51.5–60.75)	0.0022 *
anti-β2GPI IgM T1	50 (43–57)	56 (51–59)	0.0560

Anti-β2GPI IgG T0 and anti-β2GPI IgG T1 antibodies were not found to be positive in any of the subjects. Abbreviations: %–percent of patients; ACL—anti-cardiolipin antibodies; anti-β2GPI—anti-beta-2 glycoprotein I antibodies; T0—laboratory assay before vaccination; T1—laboratory assay before the third dose (the median number of days: 230); IQR—interquartile range; * The differences were statistically significant.

**Table 5 viruses-14-02655-t005:** The relationship between the presence of VAE2 and ANA T1 titres.

Severity of VAE2	Min.	1st. Qu.	Median	Mean	3rd Qu.	Max.	SD	SE
0–5 *	0	0	0	66.4	100	1000	120.2	8.9
6–10	0	0	0	93.7	100	1000	156.1	16.8
>10 *	0	0	100	139.3	100	1000	235.6	25.7

Abbreviations: VAE2–vaccine adverse event after the second dose; SD—standard deviation; SE—standard error; Min.—minimal; Max.—maximal; 1st. Qu—first quartile; 3rd Qu—third quartile; * Statistically significant differences.

## Data Availability

Not applicable.

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
