# Peer review of "The Risk of Autoimmunity Development following mRNA COVID-19 Vaccination"

_viruses, 2022, doi:10.3390/v14122655_

Round 1

Reviewer 1 Report

The study analyses the effect of vaccination against SARS-CoV2 on the development of autoimmunity understood as the presence of autoantibodies. The results are reassuring. I have some minor comments about the text:

1.  “Vaccination is the most promising way to return to "normal life." – this sentence sounds like a slogan. The aim of vaccination is not to return to "normal life" but to reduce the morbidity and mortality associated with SARS-Cov2 infection

2.“Assays of antibodies such as ANA, anti-ENA, ACL and anti-β2GPI were performed before vaccination (ANA T0; anti-ENA T0, APLA T0) and before the third dose in August–September 2021 (ANA T1; anti-ENA T1, APLA T1) (the median number of days: 230, IQR: 224-241.5). A second autoantibody assay was performed before the third dose, i.e., "booster". >>> repetition

3. “ The broad spectrum of interactions between autoimmune diseases and SARS-CoV-2 vaccination is not fully understood” >>>> Is the aim of the study to assess autoimmunity or the development of autoimmune diseases?

4. on what basis was the assessment of autoimmunity chosen after 7-9 months?

6. “In subjects, positive ANA antibody titres were found in 161 (45%) subjects at the first blood draw (prior to SARS-CoV-2 vaccination) and 165 (47%) subjects at the second blood draw 7-9 months after basic immunization.” OR “ When analyzing only patients with ANA antibody titres of 1:320 (n=39) and higher at follow-up after 7-9 months, there was an increase in the titres in only five subjects, while there was no change in the titres in 13 subjects, and 21 subjects experienced a decrease in their ANA antibody titres “ >>> 3 times “subject” in one sentence

7. Table 1 is not very readable - improve graphic form if possible

8. “ The obtained data showed no increase in the prevalence of autoimmune diseases in those vaccinated 7-9 months after complete immunization.>>> What do the authors mean by this?

9. What was the intention of dividing the study group according to the work performed? it was not clearly explained

10. “ When analyzing chronic diseases individually, i.e., cardiovascular diseases, lung diseases, neurological diseases, Hashimoto's disease, diabetes mellitus, there were no quantitative and qualitative changes with respect to ANA, anti-ENA, ACL and anti-β2GPI antibodies.>>> Why were chronic diseases analysed together and not divided into immune and non-immune diseases? It is surprising that Hashimoto's disease has not been associated with the presence of antinuclear antibodies.

Author Response

Dear Reviewer,

Thank You for taking the time to review the manuscript and for these important suggestions. We have included details of the individual comments and the proposed text corrections below.

Yours sincerely

Authors

  1. “Vaccination is the most promising way to return to "normal life." – this sentence sounds like a slogan. The aim of vaccination is not to return to "normal life" but to reduce the morbidity and mortality associated with SARS-Cov2 infection  - We fully agree that this sentence sounds like a slogan. Therefore, we changed them as suggested.

2.“Assays of antibodies such as ANA, anti-ENA, ACL and anti-β2GPI were performed before vaccination (ANA T0; anti-ENA T0, APLA T0) and before the third dose in August–September 2021 (ANA T1; anti-ENA T1, APLA T1) (the median number of days: 230, IQR: 224-241.5). A second autoantibody assay was performed before the third dose, i.e., "booster" >>> repetition - We have removed the repeated sentence.

  1. “ The broad spectrum of interactions between autoimmune diseases and SARS-CoV-2 vaccination is not fully understood” >>>> Is the aim of the study to assess autoimmunity or the development of autoimmune diseases? The study aimed to assess both the immunogenicity of vaccinations in a group of healthcare workers and to evaluate the development of autoimmune diseases after vaccination.

  1. on what basis was the assessment of autoimmunity chosen after 7-9 months? The analysis after 7-9 months was chosen because this is the period when immune reactions can already develop. In addition, it was important to get the results before the third dose of the vaccine. Due to the large study population group, it was impossible to take the blood of all of them in a specific time (one to two weeks). It was also related to the national vaccination program in our country.

  1. “In subjects, positive ANA antibody titres were found in 161 (45%) subjects at the first blood draw (prior to SARS-CoV-2 vaccination) and 165 (47%) subjects at the second blood draw 7-9 months after basic immunization.” OR “ When analyzing only patients with ANA antibody titres of 1:320 (n=39) and higher at follow-up after 7-9 months, there was an increase in the titres in only five subjects, while there was no change in the titres in 13 subjects, and 21 subjects experienced a decrease in their ANA antibody titres “ >>> 3 times “subject” in one sentence - Thank you for this comment. We have corrected both sentences.In the study population, positive ANA antibody titres were found in 161 (45%) subjects at the first blood draw (before SARS-CoV-2 vaccination) and 165 (47%) at the second blood draw 7-9 months after primary immunisation.” and “When analysing only the population with ANA antibody titres of 1:320 (n=39) and higher at follow-up after 7-9 months, there was an increase in the titres in only five subjects, while there was no change in the titres in the other 13, and 21 people experienced a decrease in their ANA antibody titres.”

  1. Table 1 is not very readable - improve graphic form if possible – We improved table 1 to make it more readable.

  1. “ The obtained data showed no increase in the prevalence of autoimmune diseases in those vaccinated 7-9 months after complete immunization.>>>What do the authors mean by this? We have not found any clinical signs that would allow us to diagnose a new autoimmune disease.

  1. What was the intention of dividing the study group according to the work performed? it was not clearly explained – The primary intention of dividing the study group according to work performed was to check whether there was significantly greater exposure of individual occupational groups (working directly with patients vs. working outside the COVID zone), and to analysis whether protection methods were properly applied.

  1. “ When analyzing chronic diseases individually, i.e., cardiovascular diseases, lung diseases, neurological diseases, Hashimoto's disease, diabetes mellitus, there were no quantitative and qualitative changes with respect to ANA, anti-ENA, ACL and anti-β2GPI antibodies.>>> Why were chronic diseases analysed together and not divided into immune and non-immune diseases? It is surprising that Hashimoto's disease has not been associated with the presence of antinuclear antibodies. The individual effects of diseases on the presence of antibodies were evaluated. No statistical significance was found in any case. Also, for us, it was surprising that we did not find a higher frequency of ANA in the group of Hashimoto's patients. However, when reviewing the literature, there are noticeable differences in their presence between different works (3-30%).

Reviewer 2 Report

The authors refer to an important clinical issue. Overall, the manuscript has been carefully prepared. However, I have some minor comments:

1) The inclusion and exclusion criteria should be specified and a flow chart should be presented.

2) All abbreviations should be explained under each table - which is missing in some tables.

3) In the case of the correlations presented in Figures 1 and 2, where R was 0.13 and 0.16, respectively - I would advise adding an annotation about the weak strength of these correlations.

Summarizing, I believe that after a slight correction, the presented manuscript can be submitted for printing.

Author Response

Dear Reviewer,

Thank You for taking the time to review the manuscript and for these important suggestions. We have included details of the individual comments and the proposed text corrections below.

Yours sincerely

Authors

Comments and Suggestions for Authors

The authors refer to an important clinical issue. Overall, the manuscript has been carefully prepared. However, I have some minor comments:

1) The inclusion and exclusion criteria should be specified and a flow chart should be presented.  We supplemented the paragraph on Material and methods with inclusion and exclusion criteria. The exact percentage of unvaccinated people is unknown because we did not obtain permission from the hospital and unvaccinated people to disclose this data. However, it should be assumed that the percentage of people vaccinated in the facility was over 90%, due to the legal obligation to vaccinate against COVID-19 among healthcare workers during the study.

2) All abbreviations should be explained under each table - which is missing in some tables.  - Thank You for this valuable comment. We have added and corrected the abbreviations under the tables.

3) In the case of the correlations presented in Figures 1 and 2, where R was 0.13 and 0.16, respectively - I would advise adding an annotation about the weak strength of these correlations. - Thank You for this comment. We added information about the weak correlation strength under the figures to better read the data.

Summarizing, I believe that after a slight correction, the presented manuscript can be submitted for printing.

Reviewer 3 Report

There is some evidence that autoimmunity prior to COVID-19 infection is associated with more severe disease.  So it is important that this study established an immunological baseline before immunization.

The authors are to congratulated on their important work.

This is a study of medical professionals who received a complete COVID-19 who were vaccinated with 2 doses of Pfizer-BioNTech's BNT162b2.

The subjects had blood drawn before and after vaccination and immunologic assays were conducted on both sets of samples, allowing a before-and-after comparison.

The literature on the fact that a huge disparity between prior antibodies autoimmunity (something like 78%) in cases of severe COVID-19 and only 7-8% in cases of mild COVID-19 is robust and might be useful to cite.

It is also evident that Th-2 skewed immune systems may not far as well with exposure to SARS-CoV-2 epitopes in vaccines or in exposure by infection.

This citation will help place the results in context.

Iverson GM, Victoria EJ, Marquis DM. Anti-beta2 glycoprotein I (beta2GPI) autoantibodies recognize an epitope on the first domain of beta2GPI. Proc Natl Acad Sci U S A. 1998 Dec 22;95(26):15542-6. doi: 10.1073/pnas.95.26.15542. PMID: 9861005; PMCID: PMC28079.

https://www.ncbi.nlm.nih.gov/pmc/articles/PMC28079/

The MS has numerous hyphens out of place.

Author Response

Dear Reviewer,

Thank You for taking the time to review the manuscript and for these important suggestions. We reviewed the proposed publication and added it to the discussion. We have corrected the numbering of references. Throughout the manuscript, we have tried to improve the hyphens.

Yours sincerely

Authors